# Pediatric Odontogenic Sinusitis: A Systematic Review

**DOI:** 10.3390/jcm13082215

**Published:** 2024-04-11

**Authors:** Cecilia Rosso, Anastasia Urbanelli, Chiara Spoldi, Giovanni Felisati, Giancarlo Pecorari, Carlotta Pipolo, Nicolò Nava, Alberto Maria Saibene

**Affiliations:** 1Otolaryngology Unit, Santi Paolo e Carlo Hospital, Department of Health Sciences, Università degli Studi di Milano, 20142 Milan, Italy; cecilia.rosso@unimi.it (C.R.); chiara.spoldi@unimi.it (C.S.); giovanni.felisati@unimi.it (G.F.); carlotta.pipolo@unimi.it (C.P.); nicolo.nava@unimi.it (N.N.); 2Otorhinolaryngology Unit, Department of Surgical Sciences, University of Turin, 10124 Turin, Italy; anastasia.urbanelli@gmail.com (A.U.); giancarlo.pecorari@unito.it (G.P.)

**Keywords:** pediatric, sinusitis, oral health, endoscopy, computed tomography, complications, antibiotics

## Abstract

**Background**: Pediatric odontogenic sinusitis (PODS) is a rare condition with limited research on its clinical features, diagnostic criteria, and treatment options. The current guidelines on pediatric rhinosinusitis do not mention a possible dental origin of the disease. This systematic review aims to summarize and analyze the existing literature on PODS, focusing on epidemiology, etiology, diagnostic tools, complications, treatment options, and outcomes. **Methods**: A systematic review was conducted following PRISMA reporting guidelines. Electronic searches were performed in multiple databases using keywords related to PODS and therapeutic strategies. Original articles reporting data on treatment outcomes for PODS were included. **Results**: The review highlighted the scarcity of high-quality evidence on PODS. The literature mainly consists of case reports and low-grade evidence studies. Limited data on the epidemiology, etiology, diagnostic tools, complications, and treatment outcomes of PODS in children are available. **Conclusions**: Further research is needed to better understand the clinical features, diagnosis, and treatment of PODS in pediatric patients. High-quality studies are required to establish evidence-based guidelines for the management of this condition, especially given the apparently high rate of complications when compared to adult ODS.

## 1. Introduction

Odontogenic sinusitis (ODS) is a prevalent condition characterized by secondary sinusitis induced by dental disease or complications arising from dental procedures. Although ODS has received less attention compared to other forms of rhinosinusitis, recent systematic reviews and international consensus statements have contributed to its increased recognition and definition [1,2,3,4,5,6]. Conversely, pediatric odontogenic sinusitis (PODS) remains understudied, lacking substantial high-quality evidence pertaining to its clinical features, diagnostic criteria, and treatment options.

ODS can be classified as a localized, unilateral, secondary chronic rhinosinusitis (CRS) [7]. The most recent consensus defines ODS as bacterial maxillary sinusitis, with or without extension to other paranasal sinuses, resulting from adjacent infectious maxillary dental pathology or complications following dental procedures [1]. Dental causes can be identified in up to 30–40% of maxillary sinusitis cases [2,7,8]. The clinical presentation of ODS shares similarities with rhinosinusitis symptoms [9], including nasal obstruction, nasal discharge, facial pain, and hyposmia. Foul smell is particularly associated with ODS [1]. Some patients may be asymptomatic, and dental pain is infrequent. Nasal endoscopy and computed tomography (CT) scans play crucial roles in diagnosing ODS. Nasal endoscopy typically reveals purulent secretions in the middle meatus, although edema and polyps may also be present. CT scans often exhibit the complete opacification of the maxillary sinus as a specific finding [10,11,12]. The disease can extend to other paranasal sinuses, especially those in the anterior compartment such as the anterior ethmoid and frontal sinuses. Collaboration with dental specialists is essential in confirming the diagnosis of ODS based on CT scans and/or consultations. Consequently, a multidisciplinary approach involving otolaryngologists, dental providers, and patients is necessary for treatment planning [3,13]. Therapeutic strategies for ODS encompass dental treatment and/or endoscopic sinus surgery (ESS) [8,14]. A combined treatment involving ESS and the closure of oroantral fistulas (OAFs) is recommended for ODS with OAFs [15]. ESS is the primary treatment for dental implant-related ODS unless peri-implantitis is present or the implant is not osteointegrated [16]. In cases where apical periodontitis is identified as the cause of ODS, dental extraction has shown the highest success rates. ESS can be considered as a secondary line of therapy if dental treatment alone proves ineffective [17].

While knowledge and management guidelines regarding ODS in adults continue to advance, the literature on PODS remains limited, predominantly comprising case reports and low-grade evidence studies. The most recent European position paper on rhinosinusitis [18] provides a clear definition of rhinosinusitis in children and an integrated care pathway for pediatric CRS treatment. However, it does not address the potential odontogenic origin of the disease, focusing instead on causes such as adenoid hypertrophy, cystic fibrosis (CF), primary ciliary dyskinesia (PCD), and primary immunodeficiency (PID). While the paper addresses thoroughly the treatment approaches for pediatric rhinosinusitis, the guidelines do not offer specific recommendations for diagnosing or treating dental pathology. Similarly, the American Academy of Pediatrics (AAP) guidelines on acute bacterial sinusitis (ABS) [19] do not address the topic of dental sources of infection, except from a microbiological perspective, specifying that the identification of anaerobes in cultures of purulent discharge from maxillary sinusitis is associated with PODS in both children and adults [20,21]. Diagnosis is typically based on symptoms and contrast-enhanced CT scans if complications are suspected. Antibiotics constitute the mainstay of therapy. ABS prevalence among children seeking care for respiratory illness is approximately 6–7%, with complications such as orbital and intracranial infections not uncommon [22]. Despite existing reports of PODS and its complications [23], the literature on the subject is mostly anecdotal, thus not allowing us to build a specific management framework accounting for pediatric needs. From a diagnostic standpoint, while CT is a routine tool for ODS diagnosis in adults, radiation exposure should be kept as low as possible in children. Thus, otolaryngological examination (especially with the use of nasal endoscopy) and dental examination might carry much more significance in identifying sinusitis and suspecting the odontogenic focus in PODS than in its adult counterpart. Again, dental etiologies in PODS are expected to be different, with a significantly lower emphasis on implantology, and a more relevant role for conservative dentistry and prevention of classic dental disease. Consequently, further study and understanding of PODS in children are warranted, especially for the management of complicated cases.

This review aims to provide a comprehensive summary and analysis of the existing literature on PODS, with a particular focus on its epidemiology, etiology, diagnostic tools, complications, treatment options, and treatment outcomes.

## 2. Materials and Methods

### 2.1. Inclusion and Exclusion Criteria

All types of articles were considered for inclusion, except for meta-analyses, systematic reviews, and narrative reviews. However, these review articles were manually checked for any potentially relevant papers. The exclusion criteria included non-human studies, papers published in languages other than English, Italian, German, French, or Spanish, studies not primarily focused on PODS, and studies lacking clear descriptions of the therapeutic strategies. There were no restrictions based on the minimum study population or the publication date.

### 2.2. Information Sources and Search Strategy

Following protocol registration in the Open Science Framework database (available at https://osf.io/tuv4y, (accessed on 25 March 2024)), we conducted a Cochrane method systematic review between 1 November 2023 and 31 January 2024, following the PRISMA reporting guidelines [24]. Systematic electronic searches were performed in English, Italian, German, French, and Spanish to identify articles reporting original data on therapeutic strategies and outcomes for Pediatric Odontogenic Sinusitis (PODS).

On 2 November 2023, we conducted a primary search on the MEDLINE (through the PubMed search engine), Embase, Web of Science, Cochrane Library, Scopus, and ClinicalTrials.gov databases. The search terms used were as follows: “(child* OR pediatr* OR infan* OR newborn* OR new-born* OR perinat* OR neonat* OR baby OR babies OR toddler* OR minors* OR boy OR boys OR girl OR girls OR kid OR kids OR preschool* OR schoolchild* OR ‘school child*’ OR adolescen* OR juvenil* OR youth* OR teen* OR underage* OR ‘under age’ OR pubescen* OR puberty OR paediatric* OR peadiatric*) AND (sinusitis OR rhinosinusitis) AND (odontogenic OR implant OR ‘dental implant’ OR tooth OR ‘sinus elevation’ OR ‘sinus augmentation’ OR ‘sinus lift’ OR ‘dental implantation’ OR fistula OR extraction OR endodontic)”. The complete search strategies and the number of items retrieved from each database are provided in Table 1. Additionally, we examined the references of selected publications to identify any further relevant reports that were not captured by the initial database search. The same selection criteria were applied to these additional reports.

### 2.3. Selection Process

Abstract and full-text reviews were conducted in duplicate by different authors. During the abstract review stage, all studies deemed eligible by at least one reviewer were included. Disagreements during the full-text review stage were resolved through consensus among the reviewers.

### 2.4. PICOS Criteria

The Population, Intervention, Comparison, Outcomes, and Study (PICOS) framework for this review was defined as follows:

P: All pediatric patients diagnosed with odontogenic sinusitis (ODS)

I: Any type of treatment for PODS, including surgical, medical, or combined approaches

C: Comparison between different types of treatments

O: Success rate of the selected treatment and the incidence of complications

S: Original studies conducted in any clinical setting, excluding meta-analyses

### 2.5. Data Extraction

For each included article, the following data were recorded: study type, total number of patients, female-to-male ratio, age of patients at diagnosis, diagnostic methods (radiological investigations, clinical evaluations such as endoscopy and blood samples), etiology, primary treatment (surgery, medical therapy, or combined), outcomes after primary treatment (response or failure), prior therapy before the final diagnosis of PODS, concurrent therapy in addition to primary treatment, subsequent treatments following the primary treatment, and follow-up period in months.

Data extraction was performed in duplicate by two authors, with discrepancies resolved through consensus.

### 2.6. Quality Assessment

The quality and methodological bias of the studies were assessed using the National Heart, Lung, and Blood Institute Study Quality Assessment Tools (NHI-SQAT) [25] for case series and cohort studies and the Joanna Briggs Institute Critical Appraisal tools (JBI-CAT) [26] for case reports. Similar to previous systematic reviews with middle-to-low evidence levels [27,28], items were rated as “good” if they fulfilled at least 80% of the criteria in the JBI-CAT or NHI-SQAT, “fair” if they fulfilled between 50% and 80% of the criteria, and “poor” if they fulfilled less than 50% of the criteria. The level of evidence for clinical studies was scored according to the Oxford Centre for Evidence-based Medicine (OCEBM) level of evidence guide [29]. Quality assessment was performed in duplicate by two authors, with discrepancies resolved through consensus.

### 2.7. Data Presentation and Synthesis Method

All studies were included in the data presentation, reported as text and in tables. No assumption was made about missing or unclear information, which was reported as missing or unclear. For all information recorded, the data were aggregated by descriptive statistics such as frequency.

Due to the substantial heterogeneity in study populations, methods, and the predominantly qualitative nature of the collected data, no initial or subsequent meta-analysis was planned or conducted, and no specific synthesis was provided, either in the whole group or as sub-groups. Given the scarce number of patients and studies included, no subgroup analysis or meta-regression was performed a posteriori, as well as no sensitivity analysis, synthesis risk of bias assessment, or certainty assessment.

## 3. Results

Initially, 1251 unique research items were identified, out of which 97 published reports underwent full-text evaluation. No additional reports were found during the reference-checking process. Eventually, a total of 20 studies published between 1945 and 2022 were included for analysis [23,30,31,32,33,34,35,36,37,38,39,40,41,42,43,44,45,46,47,48], as depicted in Figure 1.

Among the included studies, 5 articles were case series, while the remaining 15 articles were case reports. All articles were classified as level IV evidence according to the OCEBM scale. The articles were assessed based on the NHI-SQAT and JBI-CAT criteria, resulting in ratings of good (n = 16), fair (n = 2), or poor (n = 2). No significant bias related to the objectives of our systematic review was identified. Table 2 provides information on the study type, evidence level, and quality rating for all the included studies.

The 20 included studies involved a total of 41 participants, with an average age at diagnosis of 11 years (standard deviation, SD, ±4.19). The majority of the patients were male (30 males vs. 11 females). Multiple diagnostic methods were employed across the 20 selected studies, often utilizing more than one tool for each patient. Specifically, out of the 41 patients, 32 underwent computed tomography (CT) scans, 24 underwent X-rays, 7 received ophthalmologic evaluations, 4 underwent orthopantomography (OPT), blood samples were taken from 4 patients, nasal endoscopic examination was performed on 2 patients, 1 patient underwent magnetic resonance (MR) imaging combined with CT and ophthalmologic evaluation [29], and 1 patient was tested for tuberculosis due to the presence of a cutaneous fistulous tract [39]. In one paper (two cases), the selected diagnostic tool was not reported.

Regarding the etiology of periapical odontogenic disease spread (PODS), it was attributed to pulpitis in 9 cases, periodontal abscess in 9 cases, previous endodontic treatment (other than tooth extraction) in 5 cases, complications related to tooth extraction in 4 cases, the presence of an ectopic tooth in 8 cases, the coexistence of multiple supernumerary teeth in 1 case, tooth rupture (left upper incisor) in 1 case, and decayed teeth due to poor oral hygiene in 3 cases, and the etiology remained unknown in 1 case.

Table 3 presents demographic and clinical information regarding the treated patients, the diagnostic methods employed for each patient, and the etiology of each PODS case.

Prior to the primary treatment for PODS, 11 out of 41 patients received prior therapy. This included antibiotic therapy in eight cases, a combination of antibiotics, antihistamines, and intranasal corticosteroids in two cases, and endodontic treatment for an extensive carious lesion of a maxillary molar in one case [47]. Conversely, 30 patients did not receive any prior treatment. The antibiotic therapy regimens were specified in only six patients and included amoxicillin alone, vancomycin/meropenem/metronidazole, ampicillin/chloramphenicol/penicillin G, ampicillin/erythromycin/chloramphenicol, methicillin alone, and cephalotin alone.

The primary treatment for most patients (n = 18) was intranasal endoscopic surgery, followed by the Caldwell-Luc procedure (n = 8), anterior orbitotomy through Lynch incision (n = 2), and external craniotomy (n = 1). In two cases, the surgical procedure solely involved orthodontic management through tooth extraction. The surgical approach was not explained for four patients. In a particular study, two cases underwent Caldwell-Luc and anterior orbitotomy through Lynch incision, along with the closure of oroantral communication (OAC) [31]. Combined surgery was chosen as the primary treatment for six patients, involving various approaches such as the external approach to the nasal vestibule and intranasal endoscopic surgery [42], external inferior lid approach and tooth extraction [38], Caldwell-Luc procedure, tooth extraction, and external approach with inferior-medial orbital incision [30], external approach with lateral-orbital and Lynch incision, tooth extraction, and insertion of a drainage tube from the nostril (due to extensive neurological involvement with cerebral abscess) [47], Caldwell-Luc approach and craniotomy [35], and intranasal endoscopic surgery and tooth extraction [39]. Concurrent therapy was administered in 11 out of 41 patients, consisting of antibiotic therapy (n = 10) and intranasal ephedrine (n = 1). The specific therapeutic regimens for concurrent therapy were specified for six patients, including combinations such as amoxicillin/metronidazole/gentamicin, vancomycin/meropenem/metronidazole, gentamicin/clindamycin/methicillin/chloramphenicol, ampicillin/metronidazole, clindamycin/ceftriaxone, and amoxicillin/metronidazole.

Regarding the treatment response, the majority of patients (n = 35) showed a complete response to the primary therapy. Four patients experienced treatment failure with the selected approach, and for two patients, the outcome was not reported. Additional treatments were performed on two patients who responded successfully to the primary therapeutic approach. In one case, antibiotics (ampicillin and oxacillin) were administered in addition to craniotomy due to severe cerebral complications, resulting in the complete healing of the infection [35]. In the second case, oral corticosteroids (OCS) and hyperbaric oxygen therapy (HOT) were given as further treatment to a patient previously treated with surgery [30]. On the other hand, three cases received additional treatment after not responding to the primary treatment. Adenoidectomy was performed on a patient who showed failure with previous orthodontic treatment (tooth extraction) [43]. Tooth extraction was performed on a patient who did not respond to intranasal endoscopy due to PODS caused by the presence of a carious tooth [32]. Lastly, a combined approach was chosen as further treatment for a patient who did not respond to the Caldwell-Luc approach for PODS caused by an ectopic tooth. This approach involved intranasal endoscopic surgery, craniotomy for drainage of the empyema, and antibiotic therapy with metronidazole [37]. In one case, the failure of the primary treatment (anterior orbitotomy through Lynch incision) along with antibiotics (cephalotin) was not followed by further treatment due to rapid neurological deterioration and the patient’s early death [37].

Out of the 41 patients, 34 experienced complications related to PODS, with some individuals experiencing more than one complication. These complications included orbital cellulitis (n = 20), subdural empyema (n = 4), cutaneous fistulous tract (n = 2), orbital abscess (n = 1), orbital phlegmon (n = 1), seizure (n = 1), cerebritis (n = 1), cerebral abscess (n = 1), and pre-maxillary abscess (n = 1). Unfortunately, one young patient experienced a fatal outcome due to PODS [37]. On the other hand, five patients did not experience any complications, and the presence or absence of complications was not reported for two cases.

The follow-up duration was reported for 11 patients and ranged from 15 days to 2 years. Table 4 provides information on treatment regimes, outcomes, complications, and the follow-up duration for each patient included in the review.

## 4. Discussion

Pediatric odontogenic sinusitis is a rare but significant clinical condition that necessitates a comprehensive understanding of its etiology, clinical presentation, diagnostic methods, and management strategies. Acute and chronic rhinosinusitis in children accounts for approximately 2% of all annual visits to outpatient clinics and emergency departments, but there are no epidemiological studies inferring how many of these patients could actually represent misdiagnosed PODS cases. While identifying this condition in children is crucial due to its potential for severe complications and its impact on affected children’s quality of life [49], an odontogenic source is not well defined as a potential cause in the pediatric population [50] and is even not mentioned in many papers [51]. However, its detection is essential. While antibiotic therapy [50] and/or adenoidectomy [52] are the first-line treatments for general pediatric CRS, both treatments are unsatisfactory for treating PODS cases, as indicated by our review.

This systematic review revealed a poorly explored clinical condition with management that has progressively evolved alongside general clinical advancements, such as CT scans and nasal endoscopy [50].

### 4.1. Epidemiology

The analysis revealed a consistent predominance of male patients (average M:F ratio 3:1), suggesting a possible gender predisposition to this condition in the pediatric population. However, adult case series have suggested the opposite predominance [7]. The mean age of 11 years with a significant standard deviation emphasizes the importance of considering odontogenic sinusitis in the differential diagnosis of sinusitis in pediatric patients, particularly those with a history of dental infections or procedures. Dental origin is only briefly mentioned in the literature when discussing the management of both acute and chronic rhinosinusitis in children [19,53]. It is worth noting that our systematic review included only 41 patients, highlighting the rarity of this condition and, perhaps, the limited awareness surrounding it.

Facial swelling is a common presentation in pediatric patients and can have various underlying causes. Familiarity with typical clinical and imaging features, as well as the common sites of occurrence for these conditions, is crucial for establishing an accurate differential diagnosis [54]. The most frequent presenting symptoms of pediatric odontogenic sinusitis include facial pain, swelling, purulent nasal discharge, and fever, which are non-specific and can mimic other forms of sinusitis. Clinicians should maintain a high level of suspicion for odontogenic sinusitis in pediatric patients presenting with these symptoms, particularly in the context of dental infections or trauma [55].

### 4.2. Etiology

The literature lacks evidence regarding the causes of PODS since it primarily focuses on the adult population, where oroantral fistula (OAF) appears to be the most common cause, followed by apical periodontitis and periodontitis in some studies [5]. However, other case series report classic dental diseases or treatment complications without OAF as the most common cause in both pediatric and adult cases [7].

In the pediatric population, OAFs are relatively less common due to the low percentage of dental procedures during the early years of life. Therefore, it is logical that infectious diseases may be the leading cause. Indeed, our review revealed that pulpitis and periodontal abscesses are the main causes of PODS, followed by ectopic teeth and tooth extraction (see Table 3). Carious teeth, supernumerary teeth, and tooth rupture are less frequent causes of PODS.

### 4.3. Complications

Our review revealed a high rate of complications in PODS, affecting 83% of patients (see Table 4). The most common complication is orbital involvement, as the lamina papyracea remains thin in the pediatric population, allowing for the possible spread of infection to the orbit. Analysis showed that nearly half of the patients developed orbital cellulitis, while one patient had an orbital abscess. In rare cases, the infection can spread towards the intracranial fossa, with four patients presenting subdural empyema, and individual cases of cerebritis, cerebral abscess, and seizure. The literature on pediatric ODS complications is limited, with a study by Craig et al. supporting our findings. Their systematic review of complicated PODS found that 83.3% of cases involved orbital complications, while 25% were intracranial. However, their study did not differentiate between adult and pediatric populations, making direct comparisons challenging [56].

### 4.4. Diagnostic Modalities

Although clinical suspicion often leads to the correct diagnosis, imaging plays a crucial role in identifying the site of infection and potential sources of disease, such as dental abnormalities that can affect the maxillary sinus and surrounding structures [57]. Our analysis revealed that while X-rays were the primary diagnostic tool for rhinosinusitis since the 1940s [41,43], the advent of CT scans in the field of head and neck imaging in the 1980s [58] made it the leading imaging modality for diagnosing and studying ODS. The majority of patients were investigated using CT scans (see Table 3). Over the years, there has been a shift towards CT scans, with X-rays being largely replaced, aligning with current ENT guidelines [59]. CT scans provide an accurate identification of the source of infection, assessment of sinus involvement extent, and guidance for surgical planning. However, the use of ionizing radiation in CT scans raises concerns, particularly in pediatric patients, and requests for radiological examinations should be adequately supported by clinical evidence and justification [60].

PODS can lead to various complications, particularly orbital involvement, which may result in external orbital movement limitations and decreased visual acuity. Although updated guidelines on ARS/CRS do not specifically recommend routine ophthalmologic evaluation for suspected orbital involvement [18], ophthalmology consultations are highly valuable in daily practice, as ocular involvement accounts for 70% of complications in general ODS [56].

This review indicated that ophthalmologic consultations were performed when orbital involvement was suspected (see Table 3). However, in three different studies, 16 patients with orbital complications (16 orbital cellulitis cases) did not undergo ophthalmologic evaluation [32,33,38]. Although these cases were advanced abscesses, they highlight the lack of a generalized approach among otolaryngologists in managing complicated PODS cases. It is necessary to reinforce the importance of a multidisciplinary approach in the management of complicated PODS cases.

In some instances, optical coherence tomography (OCT) and blood samples were conducted to further investigate patients (four cases each, see Table 3). The limited use of these investigations may be attributed to comprehensive case studies that included CT scans and clinical examinations [8].

Nasal endoscopy, which is considered the main tool for confirming ODS based on the findings of middle meatal purulence, edema, or polyps, was rarely mentioned as a diagnostic tool. Only two patients were reported to have undergone nasal endoscopy, which contrasts with the current literature emphasizing its significance [1].

### 4.5. Treatments

The management of PODS necessitates a multidisciplinary approach involving pediatric dentists, otolaryngologists, radiologists, and infectious disease specialists. The otolaryngologist’s expertise in the subject and its potential complications is crucial in daily practice. Some authors have even proposed investigating the potential benefits of artificial intelligence in decision-making for ODS scenarios. Saibene et al. demonstrated the potential for AI to complement evidence-based clinical decision making, although there was still substantial disagreement between AI and clinical decisions, suggesting that AI is not yet optimal in assisting clinical management conclusions [61].

Our analysis revealed that antibiotics are frequently prescribed empirically as the initial management of odontogenic sinusitis to cover common pathogens. They are prescribed alone in about one in five cases or rarely in combination with antihistamines and intranasal corticosteroids (INCS). Only one case was primarily treated with endodontic treatment without success, and the majority of patients did not receive any medical treatment and were directly referred for surgery. Surgical intervention, such as endoscopic sinus surgery (ESS), Caldwell-Luc procedure, or combined approaches, is considered pivotal in ODS to achieve an adequate drainage and resolution of the infection [14]. The choice of surgical approach should be individualized based on the extent of sinus involvement, the presence of orbital complications, and the overall clinical status of the patient. All patients in our review underwent surgical treatment, with the majority undergoing an intranasal endoscopic approach (41.46%), which appears to be the most successful choice since all these patients showed treatment success. It should be noted, however, that the largest case series included [39] did not explicitly declare success in their case series, although it may be assumed based on their results. Even in cases of ODS following inflammatory processes, some authors state that the extraction of the causative tooth is an effective treatment, but a significant rate of treatment failure is reported. Other studies demonstrate that primary dental treatments for ODS have lower success rates in the range of 30–50% and emphasize the important complementary role of ESS in ODS patients [62,63].

The Caldwell-Luc approach to the maxillary sinus was the second most performed surgical approach, with only one case of failure [37]. The Caldwell-Luc approach was traditionally used for the treatment of various maxillary sinus pathologies until the introduction of endoscopic sinus surgery [15]. It is now used less frequently due to the lack of certainty regarding success rates, which can vary between 9 and 15%, consistent with our findings [64]. It is typically recommended when wider access to the sinus is required, such as for the removal of large foreign bodies [65].

A combined approach may be employed when complications arise. More than one in five patients with orbital complications underwent an inferior lid incision + tooth extraction, and in three cases, this was coupled with a Caldwell-Luc approach to drain the maxillary sinus (see Table 4). All patients who underwent the combined approach recovered from orbital complications, while one in ten non-combined treatments failed. Therefore, it can be hypothesized that a combined approach may increase healing rates when orbital involvement is present, although the majority of cases can be successfully resolved with tooth extraction and associated endoscopic sinus surgery, without the need for external incisions [66].

### 4.6. Outcomes

In our review, the overall success rate of treatment was over 80%, despite a significant proportion of patients experiencing complications such as orbital cellulitis and subdural empyema. However, it should be noted that two studies did not report the outcomes of over 20 included patients, which limits the reliability of these data. Nevertheless, these findings highlight the importance of early diagnosis and a prompt initiation of appropriate treatment to minimize the risk of serious complications.

One limitation of our review is the small number of included studies and the relatively low number of patients, which may restrict the generalizability of our findings. Future studies with larger sample sizes and longer follow-up periods are needed to further clarify the clinical course and outcomes of PODS.

### 4.7. Limitations

While the included studies provided valuable preliminary insights into pediatric odontogenic sinusitis (PODS), several limitations must be acknowledged when interpreting the results. Firstly, all studies were case reports or case series, classified as Level IV evidence according to the OCEBM scale. As such, no causal inferences could be made. Secondly, the sample sizes were small, ranging from single cases to a maximum of 18 patients in one study. This precluded robust statistical analyses. Thirdly, there was substantial clinical heterogeneity between studies in variables such as patient characteristics, diagnostic criteria, treatment protocols, and outcome definitions. This hindered direct comparisons and significantly limited the validity of the results. Furthermore, being retrospective in nature, the studies were prone to selection and reporting biases.

The weakest point remains the inconsistent use of robust diagnostic criteria based on endoscopic otolaryngological examination and the dental evaluation of the odontogenic focus. Although this shortcoming comes as expected, as the use of these criteria has been introduced only recently [1], such an inconsistent diagnostic approach significantly limits the overall conclusions of the studies and of this review.

On the other hand, our review process is partially hindered by the same heterogeneity and choice to include articles with low levels of evidence. As we expected to retrieve scarce data, we opted for providing a larger corpus of articles over their quality. 

## 5. Conclusions

In conclusion, PODS appears as a rare but potentially serious condition that requires a high level of suspicion for early diagnosis and prompt management.

Given the lack of prospective studies on the subject and the unavailability of any kind of prevalence data, the data we collected suggest maximum awareness of a potential odontogenic cause in pediatric patients with complications of sinonasal conditions. In these regards, otolaryngological and dental clinical evaluations are the standpoints of potential PODS patients, where exposure to ionizing radiation for diagnostic purposes must be reduced as much as possible. The high complication rate seems to suggest tight monitoring even in uncomplicated potential PODS cases, in order to reduce sequelae and maximize outcomes.

A multidisciplinary approach involving dentists, otolaryngologists, and other specialists is essential for achieving optimal patient outcomes. Further research is necessary to establish standardized guidelines for the management of PODS, to improve our understanding of its pathophysiology and natural history, and to understand the figures and characteristics of subpopulations at risk.

## Figures and Tables

**Figure 1 jcm-13-02215-f001:**
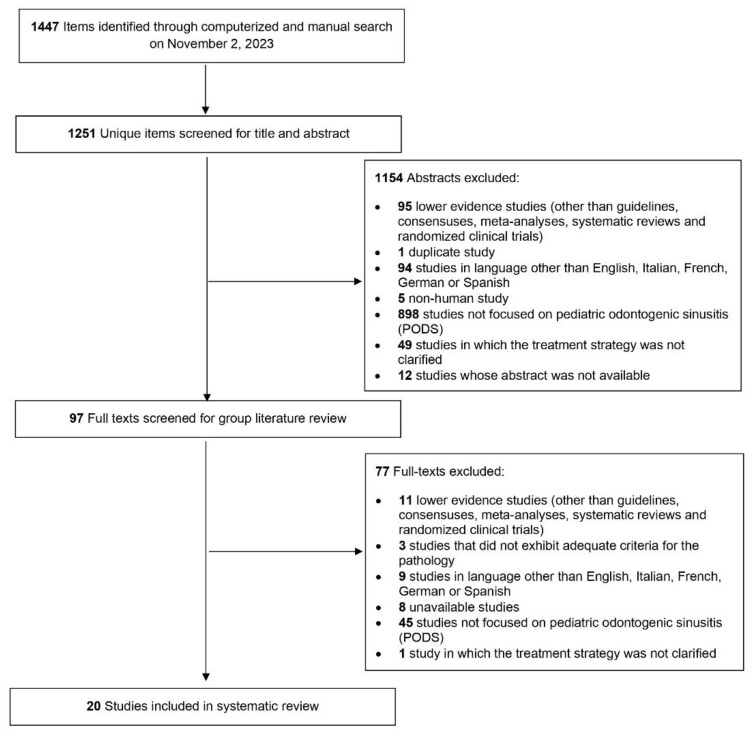
Diagram.

**Table 1 jcm-13-02215-t001:** Search strategy details and items retrieved from each consulted database.

Database	Search Date	Query	Items Retrieved (n)
Medline	2 November 2023	(child*[Title/Abstract] OR pediatr*[Title/Abstract] OR infan*[Title/Abstract] OR newborn*[Title/Abstract] OR new-born*[Title/Abstract] OR perinat*[Title/Abstract] OR neonat*[Title/Abstract] OR baby[Title/Abstract] OR babies[Title/Abstract] OR toddler*[Title/Abstract] OR minors*[Title/Abstract] OR boy[Title/Abstract] OR boys[Title/Abstract] OR girl[Title/Abstract] OR girls[Title/Abstract] OR kid[Title/Abstract] OR kids[Title/Abstract] OR preschool*[Title/Abstract] OR schoolchild*[Title/Abstract] OR “school child*”[Title/Abstract] OR adolescen*[Title/Abstract] OR juvenil*[Title/Abstract] OR youth*[Title/Abstract] OR teen*[Title/Abstract] OR underage*[Title/Abstract] OR “under age”[Title/Abstract] OR pubescen*[Title/Abstract] OR puberty[Title/Abstract] OR paediatric*[Title/Abstract] OR peadiatric*[Title/Abstract]) AND (sinusitis[Title/Abstract] OR rhinosinusitis[Title/Abstract]) AND (odontogenic[Title/Abstract] OR implant[Title/Abstract] OR “dental implant”[Title/Abstract] OR tooth[Title/Abstract] OR “sinus elevation”[Title/Abstract] OR “sinus augmentation”[Title/Abstract] OR “sinus lift”[Title/Abstract] OR “dental implantation”[Title/Abstract] OR fistula[Title/Abstract] OR extraction[Title/Abstract] OR endodontic[Title/Abstract])	67
Embase	2 November 2023	(child*:ti,ab,kw OR pediatr*:ti,ab,kw OR infan*:ti,ab,kw OR newborn*:ti,ab,kw OR ‘new born*’:ti,ab,kw OR perinat*:ti,ab,kw OR neonat*:ti,ab,kw OR baby:ti,ab,kw OR babies:ti,ab,kw OR toddler*:ti,ab,kw OR minors*:ti,ab,kw OR boy:ti,ab,kw OR boys:ti,ab,kw OR girl:ti,ab,kw OR girls:ti,ab,kw OR kid:ti,ab,kw OR kids:ti,ab,kw OR preschool*:ti,ab,kw OR schoolchild*:ti,ab,kw OR ‘school child*’:ti,ab,kw OR adolescen*:ti,ab,kw OR juvenil*:ti,ab,kw OR youth*:ti,ab,kw OR teen*:ti,ab,kw OR underage*:ti,ab,kw OR ‘under age’:ti,ab,kw OR pubescen*:ti,ab,kw OR puberty:ti,ab,kw OR paediatric*:ti,ab,kw OR peadiatric*:ti,ab,kw) AND (sinusitis:ti,ab,kw OR rhinosinusitis:ti,ab,kw) AND (odontogenic:ti,ab,kw OR implant:ti,ab,kw OR ‘dental implant’:ti,ab,kw OR tooth:ti,ab,kw OR ‘sinus elevation’:ti,ab,kw OR ‘sinus augmentation’:ti,ab,kw OR ‘sinus lift’:ti,ab,kw OR ‘dental implantation’:ti,ab,kw OR fistula:ti,ab,kw OR extraction:ti,ab,kw OR endodontic:ti,ab,kw)	92
Cochrane library	2 November 2023	((child* OR pediatr* OR infan* OR newborn* OR new-born* OR perinat* OR neonat* OR baby OR babies OR toddler* OR minors* OR boy OR boys OR girl OR girls OR kid OR kids OR preschool* OR schoolchild* OR “school child*” OR adolescen* OR juvenil* OR youth* OR teen* OR underage* OR “under age” OR pubescen* OR puberty OR paediatric* OR peadiatric*) AND (sinusitis OR rhinosinusitis) AND (odontogenic OR implant OR “dental implant” OR tooth OR “sinus elevation” OR “sinus augmentation” OR “sinus lift” OR “dental implantation” OR fistula OR extraction OR endodontic)):ti,ab,kw	256
Web Of Science	2 November 2023	TS = ((child* OR pediatr* OR infan* OR newborn* OR new-born* OR perinat* OR neonat* OR baby OR babies OR toddler* OR minors* OR boy OR boys OR girl OR girls OR kid OR kids OR preschool* OR schoolchild* OR “school child*” OR adolescen* OR juvenil* OR youth* OR teen* OR underage* OR “under age” OR pubescen* OR puberty OR paediatric* OR peadiatric*) AND (sinusitis OR rhinosinusitis) AND (odontogenic OR implant OR “dental implant” OR tooth OR “sinus elevation” OR “sinus augmentation” OR “sinus lift” OR “dental implantation” OR fistula OR extraction OR endodontic))	94
Clinicaltrials.gov	2 November 2023	(child* OR pediatr* OR infan* OR newborn* OR new-born* OR perinat* OR neonat* OR baby OR babies OR toddler* OR minors* OR boy OR boys OR girl OR girls OR kid OR kids OR preschool* OR schoolchild* OR “school child*” OR adolescen* OR juvenil* OR youth* OR teen* OR underage* OR “under age” OR pubescen* OR puberty OR paediatric* OR peadiatric*) AND (sinusitis OR rhinosinusitis) AND (odontogenic OR implant OR “dental implant” OR tooth OR “sinus elevation” OR “sinus augmentation” OR “sinus lift” OR “dental implantation” OR fistula OR extraction OR endodontic)	41
Scopus	2 November 2023	TITLE-ABS-KEY ((child* OR pediatr* OR infan* OR newborn* OR new-born* OR perinat* OR neonat* OR baby OR babies OR toddler* OR minors* OR boy OR boys OR girl OR girls OR kid OR kids OR preschool* OR schoolchild* OR “school child*” OR adolescen* OR juvenil* OR youth* OR teen* OR underage* OR “under age” OR pubescen* OR puberty OR paediatric* OR peadiatric*) AND (sinusitis OR rhinosinusitis) AND (odontogenic OR implant OR “dental implant” OR tooth OR “sinus elevation” OR “sinus augmentation” OR “sinus lift” OR “dental implantation” OR fistula OR extraction OR endodontic))	897
Total non-unique hits			1447

**Table 2 jcm-13-02215-t002:** Type of study, evidence, and quality rating of reviewed articles.

Reference	Study Type	OCEBM Rating	Quality Rating
Akhaddar et al., 2010 [23]	CR	4	F
Arunkmar, 2015 [30]	CR	4	G
Blagojeviḉ et al., 1969 [31]	CS	4	G
Blumenthal et al., 1985 [32]	CR	4	G
Brook et al., 1982 [33]	CS	4	G
Brook, 2006 [34]	CS	4	P
Brook, 2007 [35]	CS	4	P
Bullock et al., 1984 [36]	CR	4	G
Derin et al., 2015 [37]	CR	4	G
de Assis Costa et al., 2013 [38]	CR	4	G
Dhingra et al., 2015 [39]	CS	4	G
Goh, 2001 [40]	CR	4	G
Janakarajah et al., 1985 [41]	CR	4	G
Kallel et al., 2019 [42]	CR	4	F
Machado de Araujo, 1945 [43]	CR	4	G
Nisa et al., 2011 [44]	CR	4	G
Prabhu et al., 2009 [45]	CR	4	G
Ruth et al., 2022 [46]	CR	4	G
Wysluch et al., 2008 [47]	CR	4	G
Yun et al., 2015 [48]	CR	4	G

CS, case series; CR, case report; OCEBM, Oxford Centre for Evidence Based Medicine; G, good; F, fair.

**Table 3 jcm-13-02215-t003:** Demographic and clinical information on the treated patients for all included studies.

Reference	Treated Patients (n)	Female:Male Ratio (n:n)	Patients’ Mean Age at Diagnosis (Years)	Diagnosis	Aetiology
Akhaddar et al., 2010 [23]	1	1:0	11	Ophthalmologic visit, CT, MR	Endodontic treatment
Arunkmar, 2015 [30]	1	0:1	10	Ophthalmologic visit, nasal endoscopy, CT	Carious tooth
Blagojeviḉ et al., 1969 [31]	2	1:1	11.5	Ophthalmologic visit (2)	Tooth extraction (2)
Blumenthal et al., 1985 [32]	1	1:0	3	XR, CT, blood sample	Carious tooth
Brook et al., 1982 [33]	2	2:0	6.5	XR (2)	Tooth rupture; endodontic treatment
Brook, 2006 [34]	2	2:0	6.5	N/R	Periodontal abscess (2)
Brook, 2007 [35]	18	12:6	14	XR(18); CT (18)	Pulpitis (9); periodonotal abscess (7); endodontic treatment (2)
Bullock et al., 1984 [36]	1	1:0	12	CT, ophthalmologic visit	Tooth extraction
Derin et al., 2015 [37]	1	1:0	16	CT	Ectopic tooth
de Assis Costa et al., 2013 [38]	1	1:0	6	CT	Carious tooth
Dhingra et al., 2015 [39]	2	1:1	12	CT (2); TB test (1)	Ectopic tooth (2)
Goh, 2001 [40]	1	1:0	17	Nasal endoscopy, XR	Ectopic tooth
Janakarajah et al., 1985 [41]	1	1:0	14	OPT, XR, ophthalmologic visit, blood sample	N/R
Kallel et al., 2019 [42]	1	1:0	17	CT	Ectopic tooth
Machado de Araujo, 1945 [43]	1	1:0	5	Blood sample, XR	Tooth extraction
Nisa et al., 2011 [44]	1	1:0	15	CT	Ectopic tooth
Prabhu et al., 2009 [45]	1	1:0	14	OPT, CT	Ectopic tooth
Ruth et al., 2022 [46]	1	0:1	9	OPT, CT	Ectopic tooth
Wysluch et al., 2008 [47]	1	0:1	12	Ophthalmologic visit, blood sample, CT	Endodontic treatment
Yun et al., 2015 [48]	1	1:0	9	OPT, CT	Supernumerary teeth

N/R, not reported; CT, computed tomography; MR, magnetic resonance; XR, paranasal sinuses radiograph; TB, tuberculosis; OPT, orthopantomography.

**Table 4 jcm-13-02215-t004:** Treatment regimens, outcome, complications, follow-up duration.

Reference	Treated Patients (n)	Prior Therapy	Primary Treatment	Other Concurrent Treatment	Outcome	Complications	Other Further Treatments	Follow-Up (Months)
Akhaddar et al., 2010 [23]	1	Antibiotics (Amoxicillin)	Surgery: N/R	Antibiotics (Amoxicilli, Metronidazole, Gentamicin)	S	Orbital abscess	OCS, HOT	N/R
Arunkmar, 2015 [30]	1	Antibiotics (Vancomycin, Meropenem, Metronidazole)	Combined Surgery: Caldwell-Luc, tooth extraction, inferomedial-orbital incision	Antibiotics—same scheme as prior therapy	S	Orbital cellulitis	None	3
Blagojeviḉ et al., 1969 [31]	2	None	Surgery: Caldwell-Luc (1); anterior orbitotomy through Lynch incision (1); OAC closure (2)	Antibiotics	S	Orbital cellulitis (1); orbital phlegmon (1)	None	N/R
Blumenthal et al., 1985 [32]	1	Antibiotics (Ampicillin, Cloramphenicol, Penicillin G)	Surgery: intranasal endoscopic	None	F	Orbital cellulitis, seizure	Tooth extraction	12
Brook et al., 1982 [33]	2	Antibiotics (2) (Ampicillin, Erythromicin, Chloramphenicol; Methicillin)	Surgery: combined surgery with Caldwell-Luc and craniotomy (1); craniotomy (1)	None	S	Cerebritis and periorbital cellulitis (1); subdural empyema (1)	Antibiotics (2) (Ampicillin + Oxacillin)	12
Brook, 2006 [34]	2	None	Surgery (2): N/R	Antibiotics	N/R	Subdural empyema (2)	N/R	N/R
Brook, 2007 [35]	18	None	Surgery: intranasal endoscopic (16); Caldwell-Luc (2)	N/R	N/R	Orbital cellulitis (14)	N/R	N/R
Bullock et al., 1984 [36]	1	Antibiotics (Cephalotin)	Surgery: anterior orbitotomy through Lynch incision	Antibiotics (Gentamicin, Clindamycin, Methicillin, Chloramphenicol)	F	Periorbital cellulitis, cerebral abscess, death	None	N/R
Derin et al., 2015 [37]	1	Antibiotics	Combined surgery: external inferior lid approach and tooth extraction	Antibiotics (Ampicillin, Metronidazole)	S	Orbital cellulitis	None	N/R
de Assis Costa et al., 2013 [38]	1	None	Surgery: Caldwell-Luc	Antibiotics (Clindamycin, Ceftriaxone)	F	Pre-maxillary abscess, subdural empyema	Empyema’s drainage through craniotomy, antibiotics (Metronidazole)	2
Dhingra et al., 2015 [39]	2	None	Surgery: combined surgery with intranasal endoscopic surgery and tooth extraction (1); Caldwell-Luc (1)	None	S	None; fistulous tract	None	N/R; 24
Goh, 2001 [40]	1	Antibiotics	Surgery: Caldwell-Luc	None	S	None	None	N/R
Janakarajah et al., 1985 [41]	1	None	Surgery: tooth extraction	None	S	Orbital cellulitis	None	2
Kallel et al., 2019 [42]	1	None	Combined surgery: external approach to the NV and intranasal endoscopic	None	S	Fistulous tract	None	N/R
Machado de Araujo, 1945 [43]	1	None	Surgery: tooth extraction	Nasal lavage, intranasal ephedrine	F	N/R	Adenoidectomy	N/R
Nisa et al., 2011 [44]	1	None	Surgery: intranasal endoscopic	None	S	None	None	8
Prabhu et al., 2009 [45]	1	Antibiotics, systemic antihistamines, INCS	Surgery: N/R	None	S	N/R	None	0,5
Ruth et al., 2022 [46]	1	None	Surgery: Caldwell-Luc	None	S	None	None	1
Wysluch et al., 2008 [47]	1	Endodontic treatment	Combined surgery: latero-orbital and Lynch incision, tooth extraction, and insertion of drainage tube from the nostril	Antibiotics (Amoxicillin, Metronidazole)	S	Orbital cellulitis	None	N/R
Yun et al., 2015 [48]	1	Antibiotics, systemic antihistamines, INCS	Surgery: Caldwell-Luc	None	S	none	None	3

S, success; F, failure; OCS, oral corticosteroids; HOT, hyperbaric oxygen therapy; OAC, oroantral communication; INCS, intranasal corticosteroids; NV, nasal vestibule; N/R, not reported.

## Data Availability

All data pertaining to this meta-analysis are available from the authors upon reasonable request.

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
