# Peer review of "Pediatric Odontogenic Sinusitis: A Systematic Review"

_jcm, 2024, doi:10.3390/jcm13082215_

Round 1
Reviewer 1 Report
Comments and Suggestions for Authors
- The article addresses a relevant topic, highlighting the need for research on PODS in children.
- Figure 1, which should state the inclusion and exclusion criteria used to select the studies for review, is missing.
- The manuscript has two sections titled "Results".
- The authors referred to Table 5 when they described the diagnostic modalities of ODS in line 321. However, these data were presented in Table 4 instead.
- The article does not discuss the limitations of the included studies. A critical appraisal of their methodological quality is essential to assess the overall strength of the evidence.
- Given the potential variability in study designs, populations, and outcomes among the included studies, the article does not discuss or address heterogeneity. Failure to consider heterogeneity limits the validity and generalizability of the findings.
- The article briefly mentions the need for further research and evidence-based guidelines but does not provide a detailed discussion of the implications of the findings or potential clinical applications.
Reviewer 2 Report
Comments and Suggestions for Authors
Abstract:
In the abstract, refrain from mentioning the language utilized for the literature search. Please revise the abstract accordingly (Page 1, Line 23).
Keyword:
Replace the term "rhinosinusitis" with "sinusitis" if the search is exclusively for "sinusitis" (Page 1, Line 31).
Introduction:
The discussion regarding the diagnosis of sinusitis, including the use of CT scans and other aspects such as causes, needs careful consideration for its relevance to the pediatric population. It is noteworthy that The American Academy of Pediatrics, among other medical entities, advises against unnecessary exposure to radiation from medical imaging in children, promoting the use of alternative diagnostic methods when feasible, such as physical examinations, nasal endoscopy, or MRI, which does not employ ionizing radiation. In instances where a CT scan is indispensable for a child, it is imperative to minimize radiation exposure by adhering to the principle of ALARA (As Low As Reasonably Achievable). Therefore, the introduction should be amended to better address the diagnostic needs of children as opposed to adults.
The literature review indicates a gap in studies related to odontogenic sinusitis in children. It is crucial to elucidate the rationale for undertaking research in this area, given the rarity of factors leading to sinusitis in children compared to adults, such as implants and periodontitis.
The section on Page 2, Lines 81-84, appears misaligned with the pediatric context, supported by reference 24, which pertains to a case report of an 18-year-old. This section requires thorough revision to accurately reflect the condition in children rather than the adult population, which is more commonly understood.
Table 1 exclusively mentions "rhinosinusitis" rather than "sinusitis," the primary focus of the study. This requires clarification or a substantive explanation.
The manuscript should include data on the prevalence of Pediatric Odontogenic Sinusitis to underscore the significance and necessity of this research.
References:
Eliminate reference [2] due to its nature as a self-citation that does not contribute to the content substantively.
Materials and Methods:
The protocol registration details are incorrect; the start date is listed as November 2023, with an end in January 2024. Please review and update the protocol in PROSPERO accordingly.
The PRISMA guidelines applied are not the most current. Revise this section to align with the PRISMA 2020 guidelines.
Clarify the search engine utilized for the MEDLINE database inquiry.
The section must be updated to reflect the latest PRISMA guidelines comprehensively.
Discussion:
The statement “Acute and chronic rhinosinusitis in children accounts for approximately 2% of all annual visits to outpatient clinics and emergency departments” pertains to rhinosinusitis, whereas the study focuses solely on sinusitis.
Round 2
Reviewer 1 Report
Comments and Suggestions for Authors
After carefully reviewing the authors' work, I am pleased to report that they have made the necessary adjustments to address my previous comments. Upon further inspection, I have not found any additional issues that require attention.
Reviewer 2 Report
Comments and Suggestions for Authors
The author replies to each comment adequately. This article is fit for the journal's and scientific standards for publication.